# Impact of Different Diagnostic Procedures on Diagnosis, Therapy, and Outcome in Horses with Headshaking: Recommendations for Fast-Track Advanced Diagnostic and Therapeutic Protocols

**DOI:** 10.3390/ani12223125

**Published:** 2022-11-13

**Authors:** Tanja Kloock, Maren Hellige, Anke Kloock, Karsten Feige, Tobias Niebuhr

**Affiliations:** 1Clinic for Horses, University of Veterinary Medicine Hannover Foundation, Bünteweg 9, 30559 Hanover, Germany; 2Department of Cell Biology, NYU Langone Medical Center, 550 First Avenue, New York, NY 10016, USA

**Keywords:** headshaking, diagnostic investigation, computed tomography, therapy, horse

## Abstract

**Simple Summary:**

The major cause of headshaking in horses is idiopathic trigeminal-mediated headshaking, which often has a guarded long-term prognosis. Thus far, it is diagnosed by the exclusion of other differentials. Therefore, in most horses presented with headshaking, a considerable number of diagnostic procedures and different therapies are performed. The objectives of this study were to analyse the impact of different diagnostic procedures on diagnosis, therapy, and outcome in horses with headshaking and to establish recommendations for fast-track advanced diagnostic and therapeutic protocols. An underlying pathology causing the headshaking signs was identified in 6% of the studied horses. After the exclusion of differentials via history and clinical signs, most of the clinically relevant findings were detected with computed tomography, and causality was proven through positive local anaesthesia or targeted therapy. Thus, the number of diagnostic procedures can be significantly reduced. All the horses with underlying conditions demonstrated a positive outcome. In conclusion, the clinical relevance of diagnostic findings should be verified through local anaesthesia or targeted therapy, depending on the invasiveness of procedures, risks, and expected benefits. Furthermore, thorough patient history, clinical signs, and computed tomography turned out to be most beneficial and should be considered as core components in a short diagnostic protocol.

**Abstract:**

Most horses affected by headshaking (HS) are diagnosed with idiopathic trigeminal-mediated headshaking (i-TMHS) when no underlying disease is found. Diagnosis is made by the exclusion of differentials considering history, clinical signs, and diagnostic investigations. Therefore, in horses presented with headshaking, many diagnostic procedures and therapies are conducted. Retrospectively, the digital patient records of 240 horses with HS were analysed regarding the impact of diagnostic procedures on diagnosis, therapy, and outcome. Horses were extensively examined using a standardised protocol including clinical (ophthalmologic, orthopaedic, neurologic, dental) examination, blood analysis, and imaging techniques (endoscopy, radiographs, computed tomography (CT), and magnetic resonance imaging). Many findings were revealed but were of clinical relevance in only 6% of the horses. These horses were, therefore, diagnosed with secondary headshaking (s-HS). In addition, all of these horses demonstrated a positive outcome. The CT of the head revealed 9/10 of the clinically relevant findings. Other diagnostic procedures had no major additional impact. Conclusively, the diagnostic investigation of horses with HS should aim at differentiating i-TMHS from s-HS. The clinical relevance of findings should be verified through diagnostic anaesthesia or targeted therapy depending on risks, invasiveness, and expected benefits. To reduce the multitude of examinations, diagnostic investigations should focus on the CT of the head in those horses with suspicion of i-TMHS based on typical history, clinical signs, and physical examination.

## 1. Introduction

Headshaking (HS) is defined as a symptom leading to the spontaneous and uncontrolled movements of the head without an obvious external trigger [1]. Typical clinical signs are vertical HS accompanied by electric-shock-like jerking and nasal irritation [2,3,4]. Since Aleman et al. (2013, 2014) found a reduced stimulation threshold of the trigeminal nerve of affected horses [5,6], the disease is called trigeminal-mediated headshaking (TMHS) [7]. In rare cases, secondary nerve irritation is caused by different pathologies in horses with headshaking [2,5,8,9,10,11,12,13,14,15,16,17]. In contrast, in most horses, no underlying disease can be diagnosed, and it is believed to be a primary, possibly functional disorder of the trigeminal nerve [2,5,10]. Therefore, this disease is called idiopathic TMHS (i-TMHS) [10,18].

The diagnosis of i-TMHS is based on the history of the horse, clinical signs, and a thorough diagnostic investigation without any findings able to cause secondary trigeminal sensitisation [10]. Therefore, i-TMHS is diagnosed by excluding the pathologies of all the organ systems that could lead to HS. Thus far, these organ systems have thoroughly been examined via physical examination, blood analysis, ophthalmoscopy, otoscopy, oral examination, an endoscopy of the airways, radiography, computed tomography (CT), and positive diagnostic local anaesthesia of the maxillary nerves [10,19,20,21].

As the aetiology of i-TMHS remains unknown, the therapy of affected horses is challenging [19]. A great variety of different therapies and management strategies exist, such as the application of nose nets or face masks, pharmaceuticals, surgical techniques, and neurostimulation. Nevertheless, many horses do not respond to these treatment options, and therefore the prognosis for horses with i-TMHS is generally described as guarded with disease progression over time [1,7,10,18,19,22,23,24].

As a thorough and correct diagnostic investigation of headshaking horses so far has been elaborate and financially demanding, the aims of this study were to analyse the diagnostic procedures and findings in a cohort of horses with HS and their impact on diagnosis, therapy, and outcome. Based on those results, recommendations for fast-track advanced diagnostic and therapeutic protocols for horses with HS should be established. 

## 2. Materials and Methods

### 2.1. Horses

Digital patient records were analysed from horses that were referred or diagnosed with HS at the Clinic for Horses, University of Veterinary Medicine Hannover, between 2006 and 2021. The history, clinical signs, and diagnostic procedures of those horses were assessed.

### 2.2. Diagnostic Investigation

All the horses presented with HS were investigated by using the same standardised examination protocol according to university guidelines: general clinical examination, blood laboratory analysis (haematology and blood biochemistry), ophthalmologic, oral, clinical neurologic, and orthopaedic examination, including cervical and thoracolumbar spine, the endoscopy of the upper and lower airways, as well as the endoscopy of the external ear canal, radiography of the head (laterolateral and dorsoventral projections of the head as well as bilateral oblique projections of the maxillary cheek teeth), cervical and thoracolumbar spine (laterolateral projections as well as bilateral oblique projections in case of suspicious findings in laterolateral projections), the CT of the head and cervical spine, the MRI of the brain and adjacent structures, and the local anaesthesia of the maxillary nerves. If there were specific findings during the diagnostic investigation that could be clinically relevant such as severe insertional desmopathies of the nuchal ligament, those findings were diagnostically and locally anaesthetised if possible. Findings were generally graded as mild, moderate, or severe based on the existing grading systems for specific diseases (if existent) and subjective evaluation due to the clinical experience of equine specialists at the hospital.

The horses were diagnosed with i-TMHS if no pathologic finding was diagnosed that may be related to trigeminal sensitisation. Alternatively, secondary headshaking (s-HS) was diagnosed if a pathologic finding was proven to have clinical relevance for HS. Proven relevance was defined as a resolution of clinical signs following either local anaesthesia or the targeted therapy of a specific finding (Figure 1). Horses with moderate findings that were suspected not to cause trigeminal irritation or pain were diagnosed with suspected idiopathic TMHS (si-TMHS). Mild findings that are regularly seen in horses regardless of headshaking were not considered to be relevant.

In some horses, not every part of the above-mentioned diagnostic investigation was performed, because of different reasons such as poor owner compliance, risks, costs, or technical issues. No diagnosis was made if important data were missing to prove or reject the causality of a finding (such as missing information about the outcome of horses following targeted therapy of specific findings or if no diagnostic anaesthesia or targeted therapy was conducted of findings that could be causal). These patients were then categorised as ‘diagnosis not evaluable’.

### 2.3. Outcome

To evaluate the outcome of individual horses, owners were interviewed via telephone. The owners classified the outcome of clinical signs since discharge (improvement, unchanged, deterioration, or euthanasia due to HS). In horses with the seasonal appearance of clinical signs, owners were asked to compare the clinical signs of the same season throughout the years. In addition, the type of equestrian use at the time of the telephone interview was documented and compared with the time before the disease (improvement, unchanged, deterioration, or euthanasia due to HS). Furthermore, owners had to evaluate the responsiveness to therapy (improvement, no improvement) if scientifically proven therapies were conducted (medical or surgical therapy, percutaneous electrical nerve stimulation (PENS (technique according to EquiPENS^TM^)), or the targeted therapy of specific pathologic findings).

### 2.4. Data Analysis

Data were analysed using Microsoft Excel (2016) (Microsoft Corporation, Redmond, WA, USA), R Studio (Version 2022.07.1) (RStudio, PBC, Boston, MA, USA) as well as GraphPad Prism 9.0 (2020) (GraphPad Software, San Diego, CA, USA). A scoring system was applied to evaluate diagnostic findings and the outcome of horses (Appendix A). The diagnostic score was expressed in percent varying from 0% (no finding in any conducted diagnostic procedure) to 100% (severe findings in all conducted diagnostic procedures). The prognosis score is negative if horses deteriorate (−5 as the lowest possible value in case of euthanasia due to HS) and positive in cases of improvement (maximum value of 4 for resolution of clinical signs). The correlations between findings and outcomes were assessed with a Kruskal–Wallis rank-sum test [25]. A Wilcoxon rank-sum test [26], corrected for multiple testing with FDR, was used as a post hoc test. The relative abundances for response to different therapies were compared using Pearson’s chi-squared test [27].

## 3. Results

### 3.1. Horses

Between 2006 and 2021, 240 horses with HS were presented to the Clinic for Horses, University of Veterinary Medicine Hannover. Of those horses, 13 were diagnosed with other diseases than primary headshaking and were, therefore, excluded from the study (rideability issues (n = 4), behavioural disorders (n = 3), pain-related behaviour (n = 3), epilepsy (n = 2), and others (n = 1)). One horse was excluded because a neurectomy of the infraorbital nerves was already performed before presentation. Thus, 226 horses met the inclusion criteria. The study population consisted of a variety of different breeds, with a majority of warmbloods. The median age was 8 years with a range from 2 to 22 years. Geldings were over-represented (69%, 155/226) compared with mares (28%, 64/226) and stallions (3%, 7/226).

### 3.2. Diagnostic Investigation

Overall, 54% (1115/2077) of the diagnostic examinations did not reveal any pathologic findings, whereas 1% (20/2077) of the examinations revealed severe findings (Table 1 and Appendix A). Throughout the diagnostic investigation, the majority of the horses had moderate findings (73%, 160/219), whereas almost none (only 0.5% (1/219)) of the horses were without findings. Additionally, mild findings were diagnosed in 18% (39/219) of the studied horses, and severe findings in 9% (19/219) (seven horses not evaluable).

Within the laboratory blood analysis, several insignificant and mild anomalies were noticed, but there were no consistent changes throughout the study population (Appendix A). Additionally, the inflammatory markers and important electrolytes for neuron functioning such as calcium and magnesium were usually within reference ranges. Horses diagnosed with s-HS presented no anomalies of blood parameters compared with horses with i-TMHS or si-TMHS. Cerebrospinal fluid was taken from 12 horses and was without abnormal adspectory or laboratory changes in all the cases.

Comparing CT and radiographs, 56% (105/189) of the findings were diagnosed only through CT and were not visible on the radiographs (mainly dental pathologies, findings at paranasal sinuses, temporomandibular joints, and infraorbital canals). In contrast, only in four cases, radiographic findings were not confirmed through CT (Table 2).

The response to local diagnostic anaesthesia was negative in many horses (66%, 66/100), regardless of the type or location of the anaesthesia (Table 3). With several findings, an individually targeted diagnostic–therapeutic treatment regime was initiated with no improvement of clinical signs in 73% (44/60) of those cases (Table 4).

In the majority of the horses (94% (142/152)), no underlying disease was diagnosed (i-TMHS: 58% (88/152), si-TMHS: 36% (54/152)). A pathology was identified as the cause of the clinical signs in ten horses (6%, 10/152), which were, therefore, diagnosed with secondary HS (s-HS) (Table 5). The oral examinations, radiographs, CT, and MRI of the head were the only diagnostic procedures, revealing the findings that caused s-HS. Nevertheless, none of the latter diagnostic procedures revealed additional relevant information compared with CT (Table 5). In 74 horses, no diagnosis was made due to missing data.

### 3.3. Outcome

The median duration from discharge to the telephone interview was 30 months (range 1–176 months). In 38/227 cases, no outcome was obtained. Deterioration and/or euthanasia due to HS was reported for 21% (40/189) of the horses (Figure 2).

The performed therapies were without success in 68% (108/159). Comparing the different therapeutic techniques, PENS (technique according to EquiPENS^TM^) was the most successful (improvement in 44% (26/59)), even though there were no statistically significant differences between responsiveness to the different conducted therapies (Figure 3, Appendix A, *p* > 0.05).

The prognosis score was independent of the diagnostic score and the results of almost all the conducted diagnostic procedures (general clinical examination, ophthalmologic, oral, clinical neurologic, and orthopaedic examination including cervical and thoracolumbar spine, the endoscopy of the upper and lower airways as well as of the external ear canal, radiography of the head, cervical and thoracolumbar spine, computed tomography of the head and cervical spine, and the MRI of the brain and adjacent structures) (Appendix A, *p* > 0.05). Horses with moderate findings during orthopaedic examination had a significantly lower prognosis score than horses without or with mild findings during orthopaedic examination (Appendix A, *p* < 0.05). Horses with findings diagnosed with radiographs of the head had a significantly lower prognosis score, even though the post hoc test did not reveal significant differences between the individual groups (no, mild, or moderate findings) (Appendix A, *p* < 0.05).

There was no statistical significance for the prognosis score in the dependency of the different diagnoses (Appendix A, *p* < 0.05). Nevertheless, the prognosis score of all the horses affected by s-HS was positive (Figure 4).

## 4. Discussion

Thus far, TMHS is a diagnosis of exclusion, as the exact aetiopathology remains unknown. Therefore, many diagnostic investigations have been conducted in the past. The aim of this study was to analyse the data of these diagnostic procedures in a cohort of 240 horses with HS and their impact on diagnosis, therapy, and outcome.

Every diagnostic procedure comes with certain risks and requires technical equipment, time effort, and costs. Therefore, unnecessary diagnostics should be avoided. In a previous study, abnormal findings during the diagnostic investigation were present in 11/100 horses with HS, but only in 2 of those horses, clinical signs disappeared after successful therapy of the abnormal findings [18]. In the present study, many examinations (54% (1115/2077)) were without any abnormal findings, but nearly every horse had mild-to-moderate findings in at least one of the conducted diagnostic procedures. Only in 6% of all horses, findings were clinically relevant as an underlying condition for s-HS. The higher number of abnormal findings in the present study can mainly be explained by the conduction of CT, which was not used in the previously mentioned study by Lane and Mair (1987).

Based on the small number of horses affected by s-HS in the present study, it must be concluded that nearly all of the findings were incidental and without any clinical relevance regarding HS. Similar findings should be expected to be diagnosed when examining the same number of horses without HS [28]. Therefore, findings within the diagnostic investigation of horses with HS should be critically reviewed, and possible correlations with HS must be carefully verified. Indications for the targeted diagnostic treatment of specific findings should be evaluated based on invasiveness, risks, and expected benefits. In addition, it can be hypothesised that i-TMHS might be an independent and functional disorder considering the very low number of horses diagnosed with s-HS within the detailed diagnostic investigations conducted.

When comparing the different diagnostic procedures, the CT of the head detected most of the clinically relevant findings, as 9/10 of the underlying diseases in horses affected by s-HS were diagnosed through CT. When examining the equine head, CT as a three-dimensional imaging technique proved to be superior to radiographs in the diagnosis of multiple clinical conditions [29,30]. Additionally, the CT of the head was evaluated as important for diagnostic investigation in horses with HS, even though only in 4/101 horses, possible causative underlying diseases were diagnosed [31]. This was confirmed within this study, as CT was more beneficial than other imaging techniques such as radiographs, endoscopy, and MRI, which did not reveal any clinically relevant additional impact. The MRI of the brain is an expensive and rather invasive (long general anaesthesia) procedure and was without findings in 86% (96/112) of the conducted examinations in this study. In addition, the conduction of an MRI examination of the equine brain is in most cases not available. Nevertheless, it must be noticed that MRI sequences were obtained from the cerebrum and parts of the cerebellum and brainstem (including local trigeminal tracts and the trigeminal ganglion, and a part of the descending trigeminal nerve). In the future, more advanced MRI techniques than those performed to this day may be able to detect abnormalities. Until further insight into pathophysiology, MRI should, therefore, be reserved for research purposes.

Conclusively, in the future, diagnostic investigations of horses with HS should focus on CT examinations of the head (Figure 5). In order to exclude other differentials, physical examinations (general clinical examination, laboratory analysis, neurologic and ophthalmologic examination) should still be conducted, as their findings cannot be identified using CT (Figure 5). Additionally, these examinations are not invasive or time-consuming, and no specialised equipment is needed.

In a study with six horses, the movements of the head only during ridden exercise and without additional signs of nasal irritation were caused by musculoskeletal pain [32]. The signs of nasal irritation, electric-shock-like jerking, and shaking of the head independently of exercise are believed to be the clinical signs of neuropathic facial pain. Therefore, a detailed evaluation of history and clinical signs is important, and an examination of non-trigeminal innervated regions (such as orthopaedic examinations) should be limited to horses without those typical clinical signs and history of TMHS (Figure 5) [10,19,32].

Maxillary nerve blocks are proposed as a diagnostic procedure to verify facial pain in horses with TMHS, as Newton et al. (2000) reported an improvement in the clinical signs of 13/17 horses [1]. Roberts et al. (2012) found a positive result in 23/27 horses after the bilateral anaesthesia of the maxillary nerve [33], while in the present study, only 21/57 maxillary blocks were positive. As shown for maxillary nerve blocks in a cadaver study, inexperienced operators, in particular, often fail to correctly deposit the anaesthetic, which might explain the false negatives [34]. Nevertheless, in this study, nerve blocks were always performed by specialists. Furthermore, the trigeminal lesions situated further centrally to the location of a maxillary nerve block must be taken into account in negative cases. On the other hand, Aleman et al. (2014) could not diminish sensory nerve conduction by placing local anaesthetic around the maxillary foramen in one horse [6]. When considering the possible complications of this procedure in combination with the questionable validity of maxillary nerve blocks, their use in the diagnostic investigations of idiopathic trigeminal-mediated facial pain needs to be revised. The statistically significant lower prognosis score for horses with moderate findings during orthopaedic examinations is possibly due to the guarded long-term prognosis for many orthopaedic diseases in horses compared with horses without orthopaedic diseases. As for the different outcomes between horses with no, mild, or moderate findings in the radiographs of the head, the authors considered this result as not relevant due to missing significance in the differences between the groups of horses within the post hoc test.

Thus far, the literature reports mostly progressive deterioration within the outcome of the disease in many horses [1,7,18]. This is in contrast to the results of the present study, as ‘only’ 21% of the horses related to HS deteriorated or were euthanised. Overall, results are difficult to compare, as this study is the first one with a large number of horses being diagnosed with a standardised diagnostic investigation by experienced equine veterinarians and not by the owners. Additionally, PENS was introduced for horses affected by TMHS only some years ago. As PENS proves to be the most successful therapy today, in this study group, it probably positively influences the prognosis of horses and might explain the better outcome in this study than those in older studies. Considering the success rates and risks of the different therapies, the authors recommend standardised therapy in horses with chronic i-TMHS (Figure 6). In cases with suspected acute onset neuropathy, anti-inflammatory and neuroprotective treatments could be beneficial.

The resolution of clinical signs after successful therapy is reported in rare cases of horses with HS due to underlying pathologies [8,9,11,12,13,14,15,16,17]. In accordance with those reports, all the horses diagnosed with s-HS within this study demonstrated a positive outcome. Comparing the outcome of horses with s-HS with the outcome of horses with i-TMHS and si-TMHS, no statistically significant difference was found, which can easily be explained by the very small number of horses with s-HS (n = 10).

Therefore, the goal of the diagnostic investigation of horses with HS should always be to distinguish between horses with i-TMHS and horses with underlying diseases (s-HS).

This means that the clinical signs of s-HS are likely to resolve if the condition can be treated successfully. In contrast, s-HS might persist if the condition is difficult to treat, and the prognosis of the underlying disease is guarded, for example, in cases of arthrosis. Therefore, the prognosis of s-HS is dependent on the underlying disease, regardless of whether it is associated with the trigeminal nervous system. That is why the horses affected by s-HS in this study were only categorised as ‘secondary’, regardless of an association between the underlying disease and the trigeminal nervous system.

The main limitation of this study is its retrospective nature and the data collection regarding the outcome. The advantage of telephone interviews compared with questionnaires is that specific questions can be asked. It was shown in previous studies that a subjective evaluation of clinical signs, especially through non-experts, demonstrates poor reliability [35]. Additionally, the placebo effect is known to have a strong impact on the evaluation of horses with HS [36]. Nevertheless, the owners evaluated disease severity without a standardised scoring system due to the retrospective nature of the study and the lack of a precise standardised scoring system at the time of study conduction.

Another limitation is that part of the outcomes were obtained in November and December. This is a time of the year when horses with seasonal headshaking often show less severe clinical signs. Nevertheless, in those cases, the owners were asked to compare the severity of the clinical signs of the same season throughout the years.

Different scientifically proven therapies as well as additional therapies were conducted. The owners often performed a multitude of management changes during treatment, which may as well affect the severity of the clinical signs and progression of the disease. Therefore, evaluating the success of specific therapies was not without bias. Nevertheless, all the included horses were examined through a standardised diagnostic investigation conducted by equine specialists experienced with headshaking. This is in contrast to other studies, in which horses were included independently of the conducted diagnostic investigation and partly without a diagnosis of HS made by a veterinarian [2,7,24,37,38]. Standardised, prospective studies with an objective scoring system are necessary to evaluate the outcome more objectively.

## 5. Conclusions

In conclusion, performing extensive diagnostic investigations in horses with a typical history and clinical signs of TMHS needs to be revised, considering the high number of incidental findings and their low impact on diagnosis, therapy, and outcome. To exclude other differentials, history and clinical signs should be precisely evaluated and combined with physical examinations. If based on those tests, other differentials can be excluded, and i-TMHS is suspected, the advanced diagnostic investigation should focus on a CT of the head, as it detects the most clinically relevant findings. Due to the better prognosis for horses affected by s-HS, relevant diagnostic findings should be ruled out via diagnostic anaesthesia or targeted specific therapy, in consideration of their invasiveness, risks, and expected benefits.

## Figures and Tables

**Figure 1 animals-12-03125-f001:**
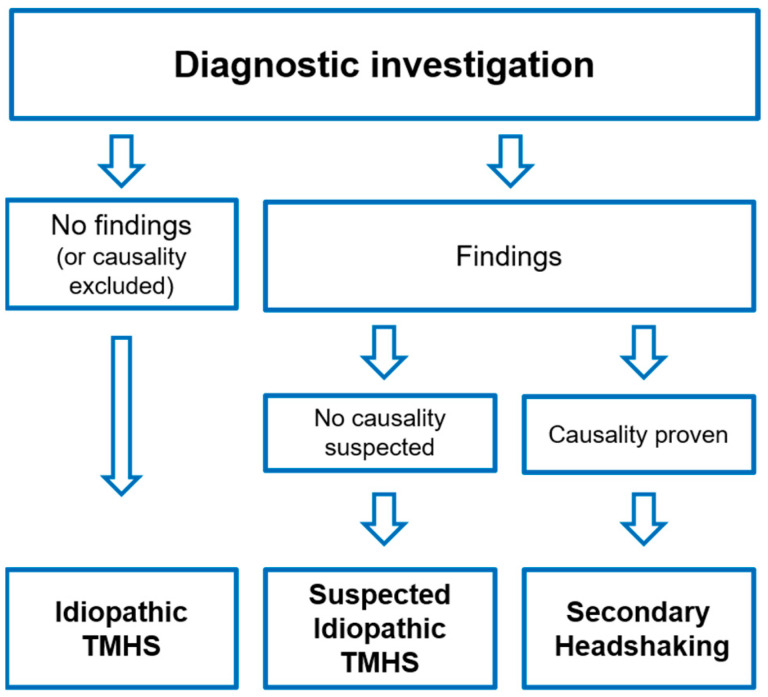
Flowchart of the diagnostic investigation of horses with headshaking in the current study. TMHS: trigeminal-mediated headshaking.

**Figure 2 animals-12-03125-f002:**
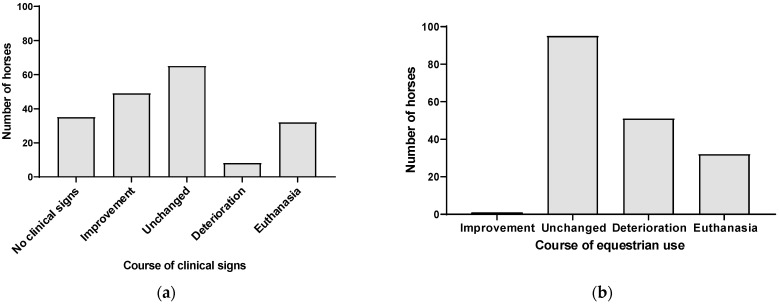
Outcome in horses with headshaking investigated in this study. Absolute abundance of outcome of clinical signs (**a**) and equestrian use (**b**) are shown. Horses with evaluable outcomes and a diagnosis of idiopathic trigeminal-mediated headshaking, suspected idiopathic trigeminal-mediated headshaking, and secondary headshaking are included.

**Figure 3 animals-12-03125-f003:**
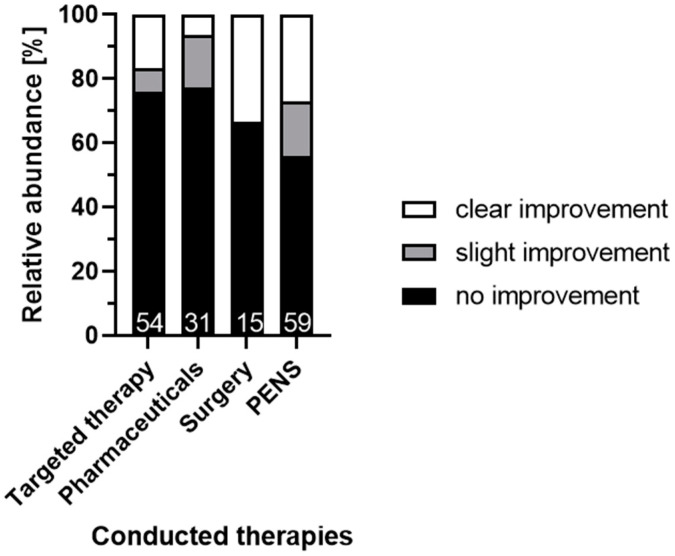
Comparison of responsiveness to different conducted therapies. Relative abundances for every therapy are shown [%]. Absolute abundances are shown at the bottom of every box (white numbers). There was no significant difference concerning the outcome between the conducted therapies (Pearson’s chi-squared test, Appendix A). Targeted therapy was conducted in case of certain suspicious findings. Pharmaceuticals included either gabapentin, carbamazepine, cyproheptadine, or corticosteroids in recommended dosages [10]. Conducted surgeries were implantation of coils in the infraorbital canals or glycerol injections within the trigeminal ganglia. PENS: percutaneous electrical nerve stimulation (technique according to EquiPENS^TM^).

**Figure 4 animals-12-03125-f004:**
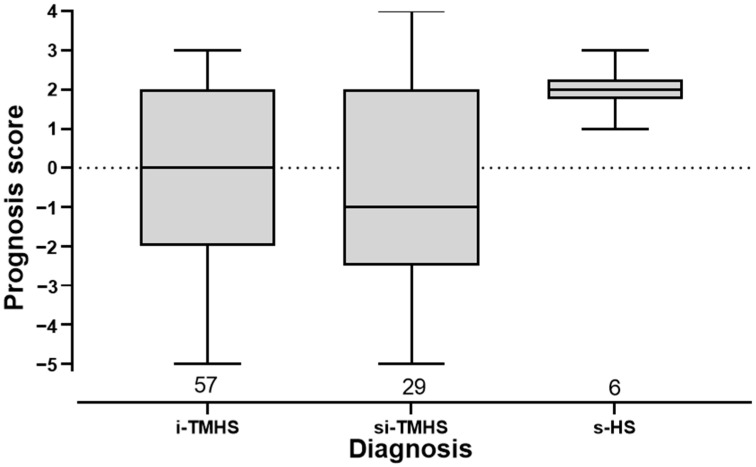
Prognosis score for different diagnoses is shown in boxplots. There were no significant differences between the different diagnoses (Kruskal–Wallis rank-sum test, *p* > 0.05). The prognosis score represents the general outcome incorporating the outcome of clinical signs, outcome of equestrian use, and responsiveness to therapy. Scores are negative for deterioration and positive for improvement (Appendix A). Horses without evaluable outcome were excluded. i-TMHS: idiopathic trigeminal-mediated headshaking, si-TMHS: suspected idiopathic trigeminal-mediated headshaking, s-HS: secondary headshaking.

**Figure 5 animals-12-03125-f005:**
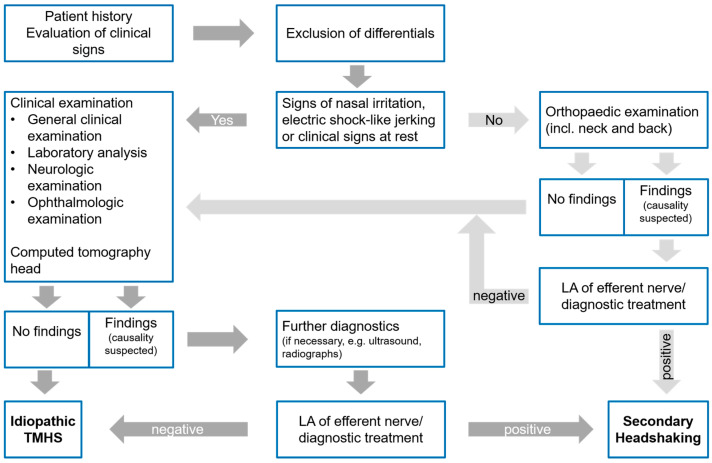
Recommendations for the diagnostic investigation in horses with clinical signs of headshaking. LA: local anaesthesia, TMHS: trigeminal-mediated headshaking.

**Figure 6 animals-12-03125-f006:**
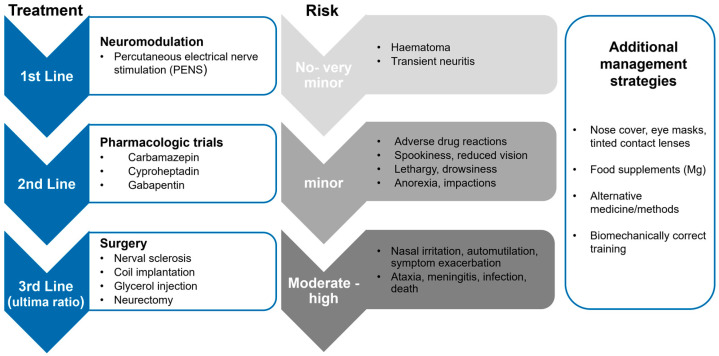
Step-wise recommendations for therapy of horses with chronic idiopathic trigeminal-mediated headshaking. First-line treatments should be applied as therapy of choice due to no to very minor risks. If neuromodulation is not effective, second-line treatments should be initiated. Surgical treatments should only be considered when euthanasia is the only option due to the invasiveness and high risk. Management strategies should be added to all conducted therapies.

**Table 1 animals-12-03125-t001:** Results of different diagnostic procedures. Relative (%) and absolute abundance (n) of findings are shown for every procedure categorised by severity. Diagnostic procedures are listed from the top to the bottom from the highest number of mild findings to the lowest. Relative abundance (%) (n); CT: computed tomography, MRI: magnetic resonance imaging.

Diagnostic Procedure	Total (n)	Without Findings	Mild Findings	Moderate Findings	Severe Findings	Not Evaluable; Not Conducted (n; n)
MRI brain and adjacent structures	112	86% (96)	6% (7)	8% (9)		1; 113
Otoscopy	133	84% (112)	12% (16)	4% (5)		0; 93
General clinical examination	225	78% (176)	18% (41)	4% (8)		0; 1
Lower airway endoscopy	182	76% (138)	24% (44)			0; 44
Radiographs cervical spine	151	74% (111)	18% (27)	9% (13)		1; 74
Clinical neurological examination	163	70% (114)	15% (24)	15% (25)		1; 62
Ophthalmological examination	168	67% (112)	19% (32)	13% (21)	2% (3)	0; 58
CT cervical spine	33	61% (20)	27% (9)	12% (4)		1; 192
Radiographs head	194	54% (105)	28% (54)	18% (34)	0.5% (1)	1; 31
Radiographs thoracolumbar spine	56	54% (30)	32% (18)	13% (7)	2% (1)	1; 169
Clinical examination of thoracolumbar spine	80	39% (31)	43% (34)	16% (13)	3% (2)	0; 146
Orthopaedic examination	63	37% (23)	35% (22)	27% (17)	2% (1)	0; 163
CT head	155	17% (27)	47% (73)	35% (55)		1; 70
Oral examination	184	6% (10)	42% (78)	47% (87)	5% (9)	3; 39
Upper airway endoscopy	178	6% (10)	62% (111)	30% (54)	2% (3)	0; 48
Total	2077	54% (1115)	28% (590)	17% (352)	1% (20)	

**Table 2 animals-12-03125-t002:** Diagnostic findings in radiographs and computed tomography (CT) for different anatomic regions. Results of both imaging techniques in the same horse were compared (only inclusion of horses with conduction of both diagnostic techniques). Negative: without finding, positive: with finding.

Anatomic Region of Finding	Radiographs and CT Negative	Radiographs and CT Positive	Radiographs Positive and CT Negative	Radiographs Negative and CT Positive	Total
Paranasal sinuses	82.39% (117)	4.23% (6)	0.70% (1)	12.68% (18)	142
*Canalis infraorbitalis*	68.31% (97)	0.70% (1)	0.00% (0)	30.99% (44)	142
Teeth	76.22% (109)	8.39% (12)	1.40% (2)	13.99% (20)	143
*Temporomandibular joint*	79.41% (54)	0.00% (0)	0.00% (0)	20.59% (14)	68
Cranial attachment of *Ligamentum nuchae*	50% (55)	46.36% (51)	0% (0)	3.63% (4)	110
Ears	96.08% (49)	0.00% (0)	0.00% (0)	3.92% (2)	51
Cervical vertebrae	54.84% (17)	32.26% (10)	3.23% (1)	9.68% (3)	31
Total	72.48% (498)	11.64% (80)	0.582% (4)	15.28% (105)	687

**Table 3 animals-12-03125-t003:** Results of different diagnostic anaesthesia procedures. Relative (%) and absolute abundance (n) of response are shown for different kinds of diagnostic anaesthesia. Relative abundance (%) (n), NSAIDs: non-steroidal anti-inflammatory drugs.

Diagnostic Anaesthesia/Result	Negative	Partly Positive	Positive
*Nervus maxillaris* (bilateral)	36/57 (63%)	13/57 (23%)	8/57 (14%)
Specific findings	30/43 (70%)	5/43 (12%)	8/43 (19%)
*Nervus maxillaris* (unilateral)	8/12	2/12	2/12
Cranial attachment of *Ligamentum nuchae*	11/12		1/12
Temporomandibular joint	3/5	2/5	
Intranasal nebulisation	2/2		
Infraorbital nerve			2/2
Infiltration at localised bone proliferation	2/2		
Systemic NSAIDs	1/2		1/2
Local infiltration of *Sinus conchae mediae*		1/1	
*Nervus mentalis*			1/1
*Nervus mandibularis*	1/1		
Cervical articular process joint (C2/C3)	1/1		
Local infiltration of the thoracolumbar spine			1/1
Temporohyoid joint	1/1		
Total	66/100 (66%)	18/100 (18%)	16/100 (16%)

**Table 4 animals-12-03125-t004:** Success of conducted targeted therapies of findings within the diagnostic investigation (n/n).

Therapy/Response	No Improvement	Slight Improvement	Notable Improvement
Teeth	24/28		4/28
Cheek tooth: extraction	10/13		3/13
Cheek tooth: shortening			1/1
Incisor tooth: extraction	6/6		
Canine tooth: extraction	1/1		
First premolar tooth: extraction	3/3		
Diastema: local therapy	4/4		
Musculoskeletal/orthopaedic diseases	7/10	1/10	2/10
Cranial attachment of the nuchal ligament: corticosteroid infiltration	1/3	1/3	1/3
Cervical articular process joint: corticosteroid injection	2/3		1/3
Distal interphalangeal joint, stifle: corticosteroid injection	2/2		
Thoracolumbar spine: corticosteroid infiltration	1/1		
Thoracolumbar spine: specific rehabilitation training	1/1		
Paranasal sinuses	3/4		1/4
Paransal sinus cyst: excision	1/2		1/2
Opening and flushing of filled *Sinus conchae medius*	2/2		
Ear	2/5	1/5	2/5
Opening and flushing *Bulla tympanica*	1/2		1/2
External ear canal: local therapy	1/3	1/3	1/3
Temporomandibular joint: corticosteroid injection	2/3		1/3
Eye	4/5	1/5	
Vitrectomy	1/2	1/2	
Enucleation	2/2		
Chromoglycin: local therapy (eye drops)	1/1		
Systemic therapy due to different diseases (including prednisolone, nonsteroidal anti- inflammatory drugs, antibiotics)	2/5	2/5	1/5
Total	44/60	5/60	11/60

**Table 5 animals-12-03125-t005:** Underlying pathologic diseases with proven causality of horses diagnosed with s-HS and the diagnostic procedures revealing the finding. CT: computed tomography, MRI: magnetic resonance imaging.

Underlying Disease	Revealing Diagnostic Procedure
Insertional desmopathy of the cranial attachment of the nuchal ligament	CT, radiographs, ultrasonography
Fracture of one *Processus paracondylaris*	CT, ultrasonography
Paranasal sinus cyst	CT, radiographs
Compression of the infraorbital canal through cheek teeth roots	CT
Sinusitis conchae mediae	CT
Otitis media sinister	CT, MRI
Muscular dysbalance in the pelvic region	Osteopathic examination
Fractured tooth root triadan 211	CT, radiographs
Partial crown fracture with pulpitis and alveolitis triadan 209	CT, oral examination
Partial crown fracture triadan 110 in combination with cyst-like lesions in the temporomandibular joint	CT, oral examination

## Data Availability

The data presented in this study are available upon request from the corresponding author. The data are not publicly available due to the inclusion of private patient data.

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
