# Peer review of "Impact of Different Diagnostic Procedures on Diagnosis, Therapy, and Outcome in Horses with Headshaking: Recommendations for Fast-Track Advanced Diagnostic and Therapeutic Protocols"

_animals, 2022, doi:10.3390/ani12223125_

Round 1

Reviewer 1 Report

Interesting and relevant study - thank you for documenting your experiences of managing this difficult disease.

I found the manusript quite difficult to follow in some areas.

On initial reading of the manuscript I found it difficult to understand exactly what you meant by "primary HS" (line 81) - it could be made a bit clearer.

Line 87-88. Can you give more information about how you evaluated the external ear, and what radiographic projections of the head and cervical and thoracolumbar spine were used?

Line 89-90. I don't understand what you mean by "specific findings if indicated".

Line 110. What was the timing of the owner follow-up? This is very important since many horses with iHS show signs that are seasonal.

Table 1. I found this difficult to follow. Why have you got two columns titled "without findings" - these show the same results. It would be helpful to move the "Total (n)" column so that it is next to the first column (Diagnostic Procedure).

Line 327. Should this read "....are likely to RESOLVE if the condition can be treated..." rather than "....are likely to RESUME if the condition can be treated...".

Author Response

Dear Reviewer 1, 

thank you very much for your very helpful comments and corrections. I adapted the manuscript according to your review report and think it should be clearer now. 

Interesting and relevant study - thank you for documenting your experiences of managing this difficult disease.

I found the manusript quite difficult to follow in some areas.

On initial reading of the manuscript I found it difficult to understand exactly what you meant by "primary HS" (line 81) - it could be made a bit clearer.

  • Thank you for this helpful comment. I rephrased the sentence and think it should be clearer now.

Line 87-88. Can you give more information about how you evaluated the external ear, and what radiographic projections of the head and cervical and thoracolumbar spine were used?

  • Thank you for this comment. I added some footnotes.

Line 89-90. I don't understand what you mean by "specific findings if indicated".

  • Thank you for this helpful comment. I rephrased the sentence and think it should be clearer now.

Line 110. What was the timing of the owner follow-up? This is very important since many horses with iHS show signs that are seasonal.

  • Thank you for this helpful comment. I added a sentence and think it should be clearer now (Line 138-140, 385-388).

Table 1. I found this difficult to follow. Why have you got two columns titled "without findings" - these show the same results. It would be helpful to move the "Total (n)" column so that it is next to the first column (Diagnostic Procedure).

  • Thank you for this helpful comment. I think there happened a mistake during adding of the table to the template, so that the column titled ‘without findings’ appeared to be duplicated.

Line 327. Should this read "....are likely to RESOLVE if the condition can be treated..." rather than "....are likely to RESUME if the condition can be treated...". 

  • Thank you very much! I changed the words!

Author Response

Dear Reviewer 2, 

thank you very much for your very helpful comments and corrections. I adapted the manuscript according to your review report and think it should be clearer now. 

Title I wonder if this better tweaked to fast-track advanced diagnostic imaging and therapeutic protocols. In a hospital a CT is pretty fast track, but not for the practitioner who has to refer and may think the title promises a quick, easy in field test with no other test required.

  • Thank you for this helpful comment. I added ‘advanced’ to the title. I did not add ‘imaging’ as the diagnostic protocol suggested includes also clinical examinations as well as careful evaluation of history and clinical signs.

Line 19 and 32: I do know what you mean but you may need to make it clear that it is horses presented for headshaking and as a major cause of headshaking is idiopathic TGM HS which is diagnosed by exclusion, horses presented for headshaking may end up having many tests. It is what you mean but turned around a little to make it clear – you don´t know it is TGMHS at presentation.

  • accepted

Line 35 - as it is a retrospective just turn the sentence around a little. Horses presented for headshaking follow a standard diagnostic plan including etc. What about if you find the diagnosis before all tests are complete? Do they still complete? You do explain this later – that they don´t - and then you can fish through the table to see what number has each but how many stop investigations after e.g. lameness exam and don´t have a CT - if you then fast tracked then just to CT, you´d mis-diagnose as TGM. So you need to do more than just bung all headshakers in the CT. I´m sure you don´t mean that but that musn´t be the take home message. You explain it a bit more in section 3.1 of the results, you are excluding those with lameness etc as a cause but there would have been tests other than CT for those so you need to emphasize the fast track advanced imaging of CT is the highest yield diagnostic procedure for those where you have a strong suspicion of TGMHS, from history and observation and physical examination at least.

  • Thank you for this helpful comment. I added a few phrases to make it clearer. (Line 20, 24-25, 35-36, 57-58, 311-314, 403-405)

Line 40- ‘CT of the head was the only procedure that revealed 9/10 of the clinically relevant find- 40 ings. Other diagnostic procedures had no major additional impact’. I think you could say CT revealed 9/10 of … etc - leave the rest the same.

  • accepted

You finish your abstract saying we should concentrate on CT for diagnostic investigations - fair enough, but somewhere you need to say how important history and observation is. I find a similar proportion of horses with secondary HS on CT - but virtually always, I suspect strongly they don´t have TGMHS before they have the scan

  • Thank you for this helpful comment. I added a few phrases to make it clearer. ( see above: Line 20, 24-25, 35-36, 57-58, 311-314, 403-405)

Line 63 - diagnostic local anaesthesia instead of blocks

  • accepted

Line 75 - again a change to advanced fast-track diagnostics

  • accepted

Line 77 - this paragraph is confusing as it suggests you only looked at TGMHS not all horses presenting for investigation of headshaking

  • Thank you for this comment. I rephrased this paragraph.

Line 83 - change diagnosed to investigated

  • accepted

Line 117 - there has been a trademark application so EquiPENSTM

  • Thank you for this comment. As we did not use the EquiPENSTM but a self-constructed device, I added the footnote ‘technique according to EquiPENSTM

Line 120 - statistical choices appear valid but are very much not my area so I would need another reviewer here

Line 148 - how do you decide if a finding is severe or moderate

  • Thank you for this comment. I added a sentence in 2.2. Diagnostic investigation (line 106-109). As there were so many diagnostic examinations conducted and many different diseases diagnosed, findings were generally graded based on subjective evaluation due to clinical experience of equine specialists at the referring hospital and if existent based on existing grading systems for specific diseases such as lameness grades.

Figure 2 is a bit confusing - do you mean TGMheadshakers - as all horses presented with headshaking

  • Thank you for this comment. I added a sentence, so it should be clearer now.

Figure 3 - what pharmaceuticals, what dose, what surgery?

  • Thank you for this comment. I added this information.

Line 239 - see also a very recently published paper by Fairburn et al in EVE

  • Thank you for this information. I added this reference (Line 289-291).

The discussion and conclusions are good.

  • Thank you very much!

The paper is good and a useful addition to the literature but emphasis throughout needs changing to make sure the reader, especially if they only read the title or abstract, fully understands that there will be more to it than ‘I have a headshaker horse‘ so ’I will do a CT’ then ‘ CT is normal therefore it’s a trigeminal mediated headshaker’.

Reviewer 3 Report

Reviewer comments for manuscript ID animals-1971575 entitled ‘Impact of different diagnostic procedures on diagnosis, therapy, and outcome in horses with headshaking: Recommendations for fast-track diagnostic and therapeutic protocols’

General comments

It is very relevant study in retrospect on an enigmatic condition of horses – Headshaking. I congratulate the authors for painstakingly putting together the clinical data for analysis as well as the clinicians for maintaining a scientific and statistically relevant clinical record of the horses affected with this syndrome. The causation of this syndrome is still an enigma, and this study also emphasized this notion. The manuscript is well written. It is precise, well organised and supported with graphs and flow charts that makes it interesting for the reader.

The authors assert that the diagnostic protocol should skip unnecessary tests/procedures. I disagree at this point as this study was not able to arrive at the specificity of the tests for the diagnosis. Hence, the conclusions have to be guarded and the scientific flow of information through the conduct of diagnostic procedures should continue till conclusive evidence of relevance is arrived at.

I have few corrections/suggestions for the authors to make before I recommend the publication of the manuscript.

Specific comments

Line 38: Please reframe ‘clinically relevant only in 6% of the horses’ as ‘but were of clinical relevance in only 6% horses’

Lines 38-40: Please rewrite this sentence. It is bit confusing for the reader.

Figure 1: Please reframe the legend as ‘Flow chart of the Diagnostic investigation of horses with headshaking’

Line 144: Please reframe ‘were without’ as ‘did not reveal’

Lines 144-49: Please clarify the criterion for categorizing the findings as ‘mild, moderate or severe’

Figure 5: Excellent flow chart. Well done.

Lines 327-29: These statements seem ambiguous. Please clarify.

Author Response

Dear Reviewer 3, 

thank you very much for your very helpful comments and corrections. I adapted the manuscript according to your review report and think it should be clearer now.

General comments

It is very relevant study in retrospect on an enigmatic condition of horses – Headshaking. I congratulate the authors for painstakingly putting together the clinical data for analysis as well as the clinicians for maintaining a scientific and statistically relevant clinical record of the horses affected with this syndrome. The causation of this syndrome is still an enigma, and this study also emphasized this notion. The manuscript is well written. It is precise, well organised and supported with graphs and flow charts that makes it interesting for the reader.

The authors assert that the diagnostic protocol should skip unnecessary tests/procedures. I disagree at this point as this study was not able to arrive at the specificity of the tests for the diagnosis. Hence, the conclusions have to be guarded and the scientific flow of information through the conduct of diagnostic procedures should continue till conclusive evidence of relevance is arrived at.

  • Thank you for your comment. I added a few phrases and sentences to emphasize the importance of evaluation of history and clinical signs in order to exclude other differentials (Line 20, 24-25, 35-36, 57-58, 311-314, 403-405). Therefore, I think it is clearer now, that we do not mean to say to put all horses with HS through CT and that´s it. I think this is what you meant, that it is important to still examine horses with headshaking very careful and precisely as it stays an enigmatic condition? In addition, I deleted the sentence ‘Other imaging techniques such as radiographs, endoscopy and otoscopy could be dispensed’ (line302). I hope with those changes the message of the manuscript is clearer now.

I have few corrections/suggestions for the authors to make before I recommend the publication of the manuscript.

Specific comments

Line 38: Please reframe ‘clinically relevant only in 6% of the horses’ as ‘but were of clinical relevance in only 6% horses’

  • accepted

Lines 38-40: Please rewrite this sentence. It is bit confusing for the reader.

  • accepted

Figure 1: Please reframe the legend as ‘Flow chart of the Diagnostic investigation of horses with headshaking’

  • accepted

Line 144: Please reframe ‘were without’ as ‘did not reveal’

  • accepted

Lines 144-49: Please clarify the criterion for categorizing the findings as ‘mild, moderate or severe’

  • Thank you for this comment. I added a sentence in 2.2. Diagnostic investigation (line 106-109). As there were so many diagnostic examinations conducted and many different diseases diagnosed, findings were generally graded based on subjective evaluation due to clinical experience of equine specialists at the referring hospital and if existent based on existing grading systems for specific diseases such as lameness grades.

Figure 5: Excellent flow chart. Well done.

  • Thank you!

Lines 327-29: These statements seem ambiguous. Please clarify

  • Thank you for this comment. By mistake, we choose the wrong word. I now changed resume through resolve. I think now it is clear what is meant with those sentences.